# RUNX3 Meets the Ubiquitin-Proteasome System in Cancer

**DOI:** 10.3390/cells12050717

**Published:** 2023-02-24

**Authors:** Albano Toska, Nikita Modi, Lin-Feng Chen

**Affiliations:** 1Department of Biochemistry, University of Illinois at Urbana-Champaign, Urbana, IL 61801, USA; 2Carl R. Woese Institute for Genomic Biology, University of Illinois at Urbana-Champaign, Urbana, IL 61801, USA

**Keywords:** E3 ligase, proteasomal degradation, RUNX3, tumor suppressor, ubiquitination

## Abstract

RUNX3 is a transcription factor with regulatory roles in cell proliferation and development. While largely characterized as a tumor suppressor, RUNX3 can also be oncogenic in certain cancers. Many factors account for the tumor suppressor function of RUNX3, which is reflected by its ability to suppress cancer cell proliferation after expression-restoration, and its inactivation in cancer cells. Ubiquitination and proteasomal degradation represent a major mechanism for the inactivation of RUNX3 and the suppression of cancer cell proliferation. On the one hand, RUNX3 has been shown to facilitate the ubiquitination and proteasomal degradation of oncogenic proteins. On the other hand, RUNX3 can be inactivated through the ubiquitin–proteasome system. This review encapsulates two facets of RUNX3 in cancer: how RUNX3 suppresses cell proliferation by facilitating the ubiquitination and proteasomal degradation of oncogenic proteins, and how RUNX3 is degraded itself through interacting RNA-, protein-, and pathogen-mediated ubiquitination and proteasomal degradation.

## 1. Introduction

Runt-related transcription factor 3 (RUNX3) is a member of the Runt domain family of nuclear transcriptional regulators in diverse biological processes, including development, cell proliferation and differentiation, senescence, DNA repair, and inflammation [1,2]. RUNX3, also called PEBP2αC, CBFA3 and AML2, contains an evolutionarily conserved Runt DNA-binding domain and a C-terminal transactivating domain [3,4,5]. RUNX3 is predominantly characterized as a tumor suppressor and is frequently inactivated in many cancers, including gastric, lung, and breast cancer [6]. However, the role of RUNX3 in cancer is more nuanced and context-dependent [7], and there is emerging evidence of the oncogenic capacity of RUNX3 in various cancers [6,8,9,10,11].

The tumor suppressor function of RUNX3 is highlighted by its transcriptional silencing from hemizygous deletion of the *Runx3* gene or promoter hypermethylation [12]. In addition, post-translational modifications play an important role in RUNX3 inactivation. RUNX3 is subject to various post-translational modifications, including acetylation, phosphorylation, methylation, and SUMOylation, that converge on the ubiquitin–proteasome system (UPS) to regulate the stability and activity of RUNX3 [13,14,15,16]. At the same time, RUNX3 exerts its tumor suppressor function by targeting oncogenic proteins for degradation via the UPS [17]. The UPS contains various components responsible for two distinct and successive steps: ubiquitination and proteasomal degradation. Ubiquitin is a small, evolutionarily conserved protein, of which the C-terminal glycine is conjugated via an isopeptide bond to distinct lysine residues on target proteins, to signal for further modification, transport, or destruction [18]. This process is usually carried out by three different enzymes, including ubiquitin-activating enzyme (E1), ubiquitin-conjugating enzyme (E2), and ubiquitin-ligating enzyme (E3), respectively. The UPS represents one fate of poly- and sometimes monoubiquitination, whereby ubiquitinated proteins are marked for degradation in the 26S proteasome [18]. Because aberrant UPS activation is implicated in cancer development [19], establishing the interactions between ubiquitination and RUNX3 in cancer could further explain the ambivalent nature of RUNX3 in carcinogenesis.

In this review, we explore two prominent interactions between RUNX3 and the UPS. First, we examine the role of RUNX3 in mediating the ubiquitination and proteasomal degradation of oncogenic proteins in cancer. Given the nature of proteins targeted by RUNX3-mediated degradation in the literature, the subsequent agency of RUNX3 as tumor suppressive is established in a cancer-specific manner. Next, we examine the cellular mechanisms and pathways that facilitate ubiquitination and proteasomal degradation of RUNX3 in cancer. We cover RNA-, protein-, and pathogen-mediated RUNX3 ubiquitination, as well as various post-translational modifications of RUNX3 that may affect its degradation in the UPS. Together, these perspectives provide a comprehensive look at RUNX3–UPS interactions, from which to gain new insights into cancer physiology.

## 2. RUNX3-Mediated Ubiquitination and Degradation of Oncogenic Proteins

### 2.1. ERα

Estrogen receptor α (ERα) is a transcription factor chiefly expressed in cells of reproductive tissue and is activated by its ligand, estrogen (E_2_), to promote cellular proliferation [20]. Abnormal estrogen signaling through Erα is associated with initiation and progression of breast cancer [21]. Overexpressed Erα is found in nearly two-thirds of breast cancers, in part due to increased protein stability [22]. As such, understanding the tight control of Erα levels would provide unique insights for the treatment approach to Erα^+^ breast cancer. 

RUNX3 functions as a tumor suppressor in breast cancer and is frequently inactivated by promoter hypermethylation and protein mislocalization, and expression of RUNX3 has been suggested as a promising prognostic biomarker in breast cancer patients [17,23]. Importantly, increased levels of RUNX3 promoter hypermethylation have been directly linked to increased Erα protein expression in Erα^+^ breast cancer [24], implying a potential role of RUNX3 in downregulating Erα protein levels. Expression of RUNX3 is inversely correlated with Erα expression in breast cancer cells and human breast cancer samples (25). Consistently, about 20% of female *Runx3^+/−^* mice spontaneously developed ductal carcinoma, with an enhanced expression of ERα and proliferation marker Ki-67 [25].

The inverse correlation between RUNX3 and ERα expression is partially due to the ability of RUNX3 to destabilize ERα through UPS recruitment [25]. RUNX3 directly interacts with ERα, and restoration of RUNX3 in MCF-7 breast cancer cells triggers the ubiquitination and degradation of ERα [25]. However, the detailed mechanism remains uncharacterized. RUNX3 binds to the hinge region of ERα, which is heavily post-translationally modified and crucial in stabilizing ERα [17]. It has been suggested that the binding of RUNX3 to the hinge domain alters its post-translational modifications, thus changing its stability [17]. It is also possible that RUNX3 facilitates the recruitment of an E3 ligase for ERα. For example, the E3 ligase, MDM2, has been shown to ubiquitinate both ERα and RUNX3 for degradation [26,27]. The binding of RUNX3 to Erα could facilitate MDM2 recruitment for Erα ubiquitination and proteasomal degradation.

### 2.2. GLI1

Hedgehog (Hh) is a conserved signaling pathway implicated in embryonic development and deregulated in certain cancers [28]. Central targets of the Hh signaling cascade are Patched receptor proteins (Ptch1 and Ptch2) and glioma-associated oncogene (Gli) transcription factors, which include GLI1, an integral transcriptional activator of downstream Hh genes [29]. A study in colorectal cancer demonstrated that increased RUNX3 protein expression resulted in both increased ubiquitination and proteasomal degradation of the oncogene GLI1, which is enhanced in the presence of suppressor of fused (SUFU), a negative regulator of GLI1 [30]. After establishing RUNX3–GLI1 interactions, it was discovered that RUNX3 recruits SKP1-CUL1-F-box (SCF), an E3 ligase super-assembly evolved to ubiquitinate F-box protein-bound substrates [31]. Specifically, the F-box protein β-TrCP was shown to ubiquitinate GLI1 [30], suggesting that the RUNX3–GLI1–SUFU trimeric complex recruits the SCF^β-TrCP^ complex to ubiquitinate GLI1 for proteasomal degradation.

### 2.3. MYCN

MYCN (N-Myc; MYCN hereafter) is a protein from a family of regulatory proto-oncogenes that include MYC (c-Myc) and MYCL1 (L-Myc) [32]. MYC family proteins regulate expression levels of nearly 15% of human genes, including those facilitating cell cycle, differentiation, and apoptosis [33]. MYCN is commonly, but not exclusively, overexpressed in neural-origin cancers, such as neuroblastomas [34]. MYCN mRNA expression has been found to increase with later stages of neuroblastoma in patient samples, while RUNX3 expression, comparatively, shows a decrease. Higher RUNX3 expression in these samples indicated a higher survival probability [35]. RUNX3 was shown not only to bind to MYCN, but also facilitate its ubiquitination and proteasomal degradation. MYCN ubiquitination increased in a dose-dependent fashion with increasing RUNX3 protein levels, suggesting that RUNX3–MYCN interaction facilitates proteasomal degradation of MYCN [35].

No E3 ligase has been directly implicated in RUNX3-mediated ubiquitination of MYCN, which is known to be ubiquitinated for proteasomal degradation by SCF^Fbw7^ [36]. In neuroblastoma, MYCN maintains stability via binding to Aurora kinase A (AURKA), which inhibits association of Fbw7 to ubiquitinate MYCN [37]. Fbw7 is a well-characterized tumor suppressor responsible for targeting many oncoproteins, such as c-MYC, Cyclin E, and mTOR, for proteasomal degradation, and it is the most commonly deregulated E3 ligase in cancer [38]. It remains to be determined whether RUNX3 similarly recruits an E3 ligase to Fbw7, to ubiquitinate MYCN. 

### 2.4. HIF-1α

Hypoxia-inducible factor 1 (HIF-1) is a transcription factor upregulated in hypoxic conditions that regulates genes for cell survivability functions, such as angiogenesis and metabolism, by binding to the hypoxia response element (HRE) on the promoters or enhancers of its target genes [39]. Cancer cells, because of their high rate of proliferation, often have increased expression of HIF-1, due to the lack of oxygen in the tumor microenvironment (TME) from poor tumor vascularity [40]. In the presence of O_2_, the alpha subunit of HIF-1 (HIF-1α) is subject to asparaginyl hydroxylation in its C-terminal transactivation domain (CTAD) by factor-inhibiting HIF (FIH) and prolyl hydroxylation by prolyl hydroxylases (PHDs) [41,42] These post-translational modifications both downregulate the transcriptional activity of HIF-1α, and facilitate its proteasomal degradation by binding to the E3 ligase von Hippel-Lindau (pVHL; hereafter VHL) [42]. Hydroxylation of Proline 564 by PHDs in the CTAD of HIF-1α is a key recognition marker for VHL to promote its ubiquitination of HIF-1α for proteasomal degradation [42].

In gastric carcinoma, RUNX3 binds to HIF-1α and PHD2, resulting in the hydroxylation of HIF-1α by PHD2, and subsequent ubiquitination of HIF-1α by VHL [43]. RUNX3 binds to PHD2, which hydroxylates proline 564 and proline 402 of HIF-1α. At the same time, RUNX3 binds to the CTAD of HIF-1α in the nucleus and facilitates the export of HIF-1α into the cytoplasm, where hydroxylated HIF-1α is recognized by VHL for ubiquitination [43]. Through this mechanism, RUNX3 may work to maintain normoxic cellular functions by facilitating O_2_-dependent hydroxylation of HIF-1α by PHD2, and subsequent UPS degradation by VHL. Recently, overexpression of miR-290 in human lung adenocarcinoma was linked to increased HIF-1α and decreased RUNX3 and PHD2 protein levels [44]. Intriguingly, HIF-1α also binds to the promoter region of miR-290 and regulates its expression, raising a possibility that there is a positive feedback loop between HIF-1α and miR-290 that might inhibit RUNX3-medated degradation of HIF-1α [43,44]. It is also interesting to note that under hypoxia, RUNX3 is silenced by G9a-mediated histone methylation and HDAC-mediated histone deacetylation on the promoters of *Runx3* in gastric cancer cells [45].

## 3. Inactivation of RUNX3 by Ubiquitination and Proteasomal Degradation

### 3.1. HOTAIR and HAGLR: RNA-Mediated RUNX3 Ubiquitination

LncRNAs, generally 200 or more base pairs in length, regulate gene expression as epigenetic and translational regulators by their influence on chromatin state or by acting as a sponge for miRNA inhibition [46,47]. In addition, a large number of lncRNAs exert their oncogenic function by affecting protein stability through direct interaction with proteins or protein complexes [47]. For example, HOX antisense intergenic RNA (HOTAIR) is a lncRNA with oncogenic properties in a variety of cancers. HOTAIR could serve as a scaffold for the Mex3b RNA-binding protein (RBP), an E3 ligase, to optimize the ubiquitination and degradation of target proteins [48].

It has been shown that HOTAIR interacts with RUNX3 via a fragment of HOTAIR spanning 1951–2100 bp and decreases the expression of RUNX3 [49]. Mechanistically, HOTAIR complexes with Mex3b to ubiquitinate and degrade RUNX3 in gastric cancer cells, leading to increased cancer invasiveness [49]. RUNX3, as a transcriptional activator of tight-junction protein Claudin1, can prevent the epithelial–mesenchymal transition (EMT) [50]. HOTAIR, through promoting Mex3b-dependent ubiquitination and degradation of RUNX3, suppresses Claudin1 expression to foster gastric cancer cell invasion and EMT [49]. In this regard, HOTAIR/Mex3b promotes gastric cancer tumorigenesis by inhibiting the RUNX3–Claudin1 tumor suppressive axis.

Recently, the Homeobox D gene cluster antisense growth-associated lncRNA (HAGLR) has been demonstrated to destabilize RUNX3 in Treg cells and regulate Treg cell differentiation [51]. Expression levels of HAGLR are directly correlated with ubiquitination of RUNX3 and inversely correlated with RUNX3 protein stability [51]. Considering that increased expression of HAGLR is linked to the progression of colon, hepatocellular, and triple negative breast cancer, where RUNX3 is often downregulated [52,53,54], it would be of great interest to determine whether a direct interaction between HAGLR and RUNX3 exists and whether HAGLR could serve as scaffold for an E3 ligase, similar to HOTAIR/Mex3b, to degrade RUNX3 via the UPS in cancer cells.

### 3.2. PIN1: Protein-Mediated RUNX3 Ubiquitination

PIN1 is a peptidyl-prolyl cis-trans isomerase (PPIase), and specifically recognizes phosphoserine or phosphothreonine residues preceding proline (pSer/Thr–Pro) motifs, and induces protein conformational changes by isomerization [55]. After binding to the pSer/Thr–Pro motif on a target protein via its N-terminal WW domain, PIN1 catalyzes cis/trans isomerization of the peptide bond via its C-terminal PPIase domain [55]. PIN1 is overexpressed in cancer and regulates numerous cancer-driving pathways by controlling the stability of oncogenes and tumor suppressors [55]. 

RUNX3 is one of the tumor suppressors of which the stability is regulated by PIN1 (59). The PIN1 WW domain binds to four pSer/Thr-Pro residues (T209, T212, T231 and S214) on RUNX3, resulting in the ubiquitination and degradation of RUNX3 in breast cancer [56]. The four PIN1 binding motifs are located immediately C-terminal of the Runt domain, which has been shown to be important for RUNX3 stability [57]. The binding of PIN1 to these phosphorylated motifs might induce the cis-trans isomerization and isomerization-mediated conformational change of RUNX3, leading to an increased accessibility of a RUNX3 E3 ligase. Further investigation is warranted to determine the E3 ligase involved in the PIN1-mediated degradation of RUNX3. PIN1 downregulates Smad2/3 proteins in the TGF-β pathway, in part through recruitment of the Smurf2 E3 ligase, to degrade these proteins [58]. RUNX3, also a target of Smurf2 in the UPS, acts synergistically with Smad proteins to activate downstream genes in the TGF-β pathway [57,59]. It has been speculated that the ubiquitination and degradation of RUNX3 by PIN1 may be carried out by Smurf E3 ligases, which also contain WW domains and degrade RUNX3 [56,57]. Furthermore, some of the known oncoprotein targets of RUNX3-mediated ubiquitination and degradation, including HIF-1α, ERα, and GLI1, are PIN1 substrates [55]. However, it remains unclear whether PIN1 is a factor of their regulation. It is possible that PIN1 might be directly or indirectly involved in the tumor suppressor function of RUNX3, and the regulation could be cell-type- or cancer-type-specific. 

### 3.3. JAB1: Nuclear Export and Proteasomal Degradation of RUNX3

c-Jun activation domain-binding protein-1 (JAB1), also known as subunit 5 of the COP9 signalosome (CSN), is a multifunctional protein that modulates signal transduction, gene transcription, and protein stability in cells [60]. Jab1/CSN5 is overexpressed in different types of cancer, and its overexpression has been implicated in the initiation and progression of many cancers [61]. Jab1/CSN5 could regulate cell proliferation by promoting the nuclear export and the degradation of several tumor suppressor proteins, including p53, p27^kip1^ and Smad4 [61].

RUNX3 is another tumor suppressor of which the activity is regulated by JAB1. JAB1 has been shown to facilitate CSN-mediated proteasomal degradation of RUNX3 in gastric cancer cells [62]. This process is accomplished by two integral interactions: the Mpr1/Pad1 N-terminal (MPN) domain of JAB1 binds to the Runt domain of RUNX3, while the nuclear export signal (NES) of JAB1 recruits the exportin, CRM1, to facilitate the export of RUNX3-JAB1 into the cytoplasm. Subsequently, the JAB1–RUNX3 complex is recruited to CSN, where the CSN-associated kinase, CK2α, phosphorylates RUNX3 and triggers phosphorylation-dependent proteasomal degradation of RUNX3 [62].

The CSN complex generally functions to reverse neddylation (rubylation) by removing NEDD8 (RUB1) from the cullin subunit of the cullin-RING-type E3 ligases (CRLs). This deneddylation stops the polyubiquitination process and allows the subsequent degradation of polyubiquitinated substrates by the proteasome [63]. JAB1 might utilize a similar mechanism for the polyubiquitination of RUNX3, but the relevant CRLs for RUNX3 remain to be determined. Further investigation may also determine whether JAB1 could directly regulate the neddylation and ubiquitination status of RUNX3 in the CSN for its proteasomal degradation.

### 3.4. H. pylori CagA: Pathogen-Mediated RUNX3 Ubiquitination

*Helicobacter pylori* is a Gram-negative bacterial pathogen that infects the gastric epithelium and causes gastritis in humans [64]. Infection by *H. pylori* presents the highest risk factor for the development of gastric cancer [64]. The carcinogenic nature of *H. pylori* is attributed to its virulence factors, notably cytotoxin-associated gene A (CagA) located in the cag pathogenicity island (cagPAI) [64,65]. *H. pylori* uses a type IV secretion system (T4SS), encoded by the cagPAI, to inject CagA directly into epithelial cells [64,65]. CagA contributes to oncogenesis by disrupting signaling pathways involved in cell shape and adhesion [64,65]. Infections with CagA^+^
*H. pylori* are linked both to increased inflammation and risk of gastric cancer [64,65].

*H. pylori* infection contributes to reduced RUNX3 expression by increasing *Runx3* promoter methylation independent of CagA, or activating the oncogenic Ras GTPase by SRCK-phosphorylated CagA [12,65]. In addition to transcriptional level regulation, *H. pylori* could also regulate RUNX3 activity via protein stability. *H. pylori* CagA binds directly to RUNX3 and promotes its ubiquitination and proteasomal degradation in gastric cancer cells [66]. While CagA relies on two N-terminal WW domains (WW1-2) to degrade RUNX3, only WW2 is required for successful binding to RUNX3 on its PPxY (PY) motif [66]. The WW1 domain could recruit an E3 ligase for the ubiquitination of RUNX3 with CagA as a scaffold protein. 

While the E3 ligase involved in CagA-mediated ubiquitination and degradation of RUNX3 remains uncharacterized, several candidate E3 ligases emerge. Smurfs canonically ubiquitinate Smads in the TGF-β pathway for degradation, while CagA induces inflammation via binding and inactivating Smads [57,67]. Conversely, RUNX3 binds and stabilizes Smads to promote activation of TGF-β downstream genes [57]. An interplay between CagA, Smurf1/2, and RUNX3 could suggest that CagA recruits Smurfs to ubiquitinate RUNX3 in gastric cancer. Furthermore, CagA-activated AKT1 in gastric cancer can also phosphorylate HDM2 (MDM2) to ubiquitinate p53 for degradation [68,69]. In addition, complex formation of CagA and CD44 in gastric cancer induces AKT-dependent activation of the Wnt pathway, from which Wnt2 is an inhibitor of p14^ARF^ [70]. Therefore, CagA–AKT signaling may induce both the activity of MDM2 and prevent its inactivation by p14^ARF^, an inhibitor of MDM2 [71]. Since RUNX3 can be shuttled into the cytoplasm and degraded by MDM2 [27], the CagA-mediated degradation of RUNX3 in the UPS may also occur through MDM2.

## 4. Regulation of RUNX3 Stability by Other Post-Translational Modifications

Several post-translational modifications can regulate the stability of RUNX3. Acetylation is one such post-translational modification that has been demonstrated to promote the stability of RUNX3. The acetyltransferase p300 binds to RUNX3 and protects RUNX3 from ubiquitination-mediated degradation via acetylation [57]. RUNX3 is acetylated by p300 on three lysine residues, K148, K186, and K192 [57]. Of note, K148 is a known target of both MDM2-mediated ubiquitination and PIAS-1 mediated SUMOylation [16,27]. Conversely, histone deacetylase 1 (HDAC1) can remove these protective modifications and facilitate nuclear export and cytoplasmic degradation of RUNX3 [15,57]. The reversible acetylation of RUNX3 by p300 and HDAC1 may serve to maintain RUNX3 stability and transcriptional activity at optimal levels. In gastric cancer cells, the histone methyltransferase (HMT) G9a methylates lysines of the Runt domain of RUNX3 (K129 and K171) to promote nuclear export and cytoplasmic degradation [15]. G9a-mediated methylation of RUNX3 inhibits the RUNX3 transactivation activity by preventing its association with CBFβ/PEBP2β subunits and p300 [15]. Thus, RUNX3 can be destabilized via Runt domain lysine post-translational modification in two ways: reversal of p300 acetylation by HDAC1 [57] and methylation by G9a [15].

Besides acetylation and methylation, phosphorylation also mediates the stability and subcellular localization of RUNX3. Phosphorylation of JAB1-exported RUNX3 in the CSN by CSKα is a necessary step for JAB1-mediated degradation of RUNX3 [62]. While export of RUNX3 into the cytoplasm is well-described in JAB1-medated degradation of RUNX3 and RUNX3-mediated degradation of HIF-1α [43,62], the way RUNX3 is re-localized to the cytoplasm preceding UPS degradation is poorly understood. SRC family kinases (SRCKs) have been shown to bind to the Runt domain and phosphorylate tyrosine residues on RUNX3 [72]. Furthermore, phosphorylation of four Ser/Thr residues (S149, T151, T153, and T155) by the PIM1 kinase on the Runt domain of RUNX3 in cancer has also been demonstrated [73]. Phosphorylation of RUNX3 may be a prerequisite for the cytoplasmic re-localization of RUNX3 and the subsequent ubiquitination and proteasomal degradation in the cytoplasm. Additionally, phosphorylation may provide unique docking sites for the recruitment of proteins involved in the ubiquitination and degradation of RUNX3. For example, the phosphorylated PY motif of RUNX3 is recognized by PIN1 [56], CagA [66], and the E3 ligases, Smurf1/2 [57]. Similarly, CDK4-mediated phosphorylation of S356 could also provide a binding motif for some UPS proteins for the ubiquitination and degradation of RUNX3 [74]. This suggests that cancer cells may overexpress kinases to phosphorylate RUNX3 at PY motifs or other target residues that, after cytoplasmic localization, could be recognized by WW domain-containing E3 ligases, such as Smurf1/2, for ubiquitination and proteasomal degradation.

SUMOylation by small ubiquitin-like modifiers (SUMOs) is a class of post-translational modification, with roles in altering protein activity, stability, and cellular localization. [75]. Much crosstalk exists between ubiquitination and SUMOylation, as SUMO ligases are related to RING domain-containing ubiquitin ligases and may even share substrates [76]. The SUMO E3 ligase, PIAS1, has been shown to bind to the Runt domain and SUMOylate K148 of RUNX3, leading to the decreased transcriptional activity of RUNX3 [16]. As K94/K148 residues are also targets for MDM2-mediated degradation of RUNX3, SUMOylation may prevent the binding and degradation of RUNX3 by MDM2 [27]. However, whether the reduced activity of RUNX3 is related to altered protein stability is still unclear.

Taken together, RUNX3 stability appears to depend upon several post-translational modifications that can either promote RUNX3 stability or destabilize it for cytoplasmic export and degradation by the UPS. It remains an interesting topic whether these different modifications occur sequentially or antagonistically to regulate the stability and tumor suppressor function of RUNX3. 

## 5. Potentials of Targeting RUNX3–UPS Interplay in Cancer Therapy

RUNX3 has been shown to be a prognostic biomarker for a variety of cancers, including breast, gastric, colorectal, and ovarian cancer, and there is a negative correlation of RUNX3 inactivation and patient survival [23,77,78]. Loss of RUNX3 expression in patients, either through hypermethylation or cytoplasmic mislocalization, correlates with poor outcomes [79,80,81,82]. RUNX3 has been suggested as a potential therapeutic target for certain cancers, since restoration of RUNX3 expression by reactivation of *Runx3* transcription or by overexpression, suppresses the proliferation of cancer cells [25,83,84]. For example, treatment of cancer cells with small molecules targeting HDACs or DNMTs can restore the expression of RUNX3 (79,80). 

However, therapeutic reactivation of RUNX3 expression through epigenetic means may be insufficient to sustain RUNX3 expression as cancer progresses, as many UPS components, including E3 ligases such as MDM2 and Smurfs, are hyperactivated in cancer cells. Thus, targeting the UPS may be effective as an adjuvant therapy for alleviating RUNX3 destabilization in cancer. Small molecules targeting various subunits of the 26S proteasome, such as bortezomib and carfilzomib, are already in clinical use to treat multiple myeloma (MM). These proteasome inhibitors are thought to shift the cellular protein equilibrium in favor of tumor suppressive and pro-apoptotic protein stability, resulting in the reduction of cancer cell proliferation [85]. Proteasomal inhibition may rescue RUNX3 from degradation, but it may also rescue oncoproteins targeted for RUNX3-mediated UPS degradation. Since the proteasome regulates the fast turnover of both p53 and p27^kip1^ [86], increased p53 stability from proteasomal inhibition may also lead to increased RUNX3 stability through p53-mediated inhibition of MDM2 [87]. High p53 levels are correlated with increased expression of p300, which facilitates acetylation-dependent stabilization of both p53 and RUNX3 [57,88]. 

Inhibitors of MDM2 and MDM2-p53 binding are in active development [89]. The therapeutic potential of maintaining MDM2-p53 homeostasis may be bifold in sustaining RUNX3 stability. In addition to blocking MDM2-mediated degradation of RUNX3, the higher resulting levels of p53 may also sustain RUNX3 in a tumor suppressive state, as p53 dysregulation is likely a switch for RUNX3 to become oncogenic [10,11]. JAB1/CSN5 inhibition may present another approach for targeting the UPS to restore RUNX3 protein expression. Several reports show a correlation between poor prognosis and an increase in JAB1/CSN5 in cancers such as pancreatic cancer, oral squamous cell cancer, and breast cancer [90,91,92]. Targeting JAB1 and the CSN may increase RUNX3 expression by preventing mislocalization and degradation of RUNX3. Recently, the NEDD8-activating enzyme (NAE) inhibitor, MLN4924, caused inactivation of CRLs and suppression of renal cell carcinoma proliferation in vitro [93]. CRL inactivation through NAE inhibitors could provide a treatment avenue for inhibiting CSN-mediated degradation of RUNX3. However, CRLs are also present in SCF complex E3 ligases, which are recruited by RUNX3 to degrade oncoproteins GLI1, and likely MYCN [30,35], so potential cancer treatments may require a more personalized approach. To achieve the best outcome in cancer therapy, a combination of small molecules targeting both epigenetic pathways and the UPS may be ideal to maintain the optimal levels of RUNX3 to sustain its tumor suppressor function.

As mentioned above, post-translational modifications of RUNX3 also play a critical role in the stability of RUNX3. Directly targeting the modifications or indirectly targeting the enzymes could also affect the stability of RUNX3 with therapeutic outcomes. For example, a recent study demonstrated that a small peptide, RMR, inhibits phosphorylation of RUNX3 at T209 by PAK1, effectively suppressing cancer cell proliferation and cancer formation [94]. Since pT209 is one of the sites recognized by PIN1 for the degradation of RUNX3 [56], it would be interesting to evaluate whether the therapeutic effect of the peptides would be partially due to the prevention of PIN1-mediated degradation of RUNX3. In a different study, G9a inhibitor, UNC0638, was found to increase apoptosis in MYCN-amplified neuroblastoma cells [85]. Given that G9a is responsible for methylating and destabilizing RUNX3 [15,45], and that RUNX3 mediates UPS degradation of MYCN [35], the anticancer properties of UNC0638 may be in part due to the increasing stability of RUNX3 for MYCN degradation. 

## 6. Concluding Remarks and Perspectives

RUNX3, a transcription factor with tumor suppressor activity, regulates multiple cellular responses, including transcription, signal transduction and cell proliferation [1,2]. The UPS is also well known to play an important role in cancer development by regulating protein function, signal transduction, transcription and apoptosis via proteolysis [18,19]. RUNX3 utilizes the UPS to exert its tumor suppressor function by inducing the degradation of oncogenic proteins (Figure 1). Additionally, RUNX3 could be inactivated via proteasomal degradation by the UPS in cancer cells (Table 1). While there is a clear interplay between RUNX3 and the UPS in cancer, which factors or signals determine whether RUNX3 is a UPS target or an activator of UPS to target other proteins is still unknown. Nevertheless, this switch likely occurs at the early stage of cancer development, when cells express normal levels of RUNX3 to target oncogenic proteins [17]. As cancer progresses, the UPS is activated with increased activity or accessibility to RUNX3, leading to the degradation and removal of RUNX3 (Figure 2). During this process, post-translational modifications of RUNX3 might be a key factor to switch RUNX3 from a UPS activator to a UPS target, as many enzymes modifying RUNX3 are overexpressed or hyperactive in cancer. Inactivation of RUNX3 by the UPS is likely an important step prior to the permanent epigenetic silencing of RUNX3 (Figure 2).

Ubiquitination and proteasomal degradation of RUNX3 represent one of the mechanisms for the inactivation of RUNX3 in the early stages of cancer development; however, the identities of a majority of these RUNX3 E3 ligases are unknown (Table 1). There are three major types of E3 ligases: the RING (really interesting new gene) family, the HECT (homologous to the E6-AP carboxyl terminus) family, and the RBR (ring between ring fingers) family [95]. E3 ligases from the RING and HECT family, such as MDM2 and Smurf1/2, have been reported to ubiquitinate RUNX3 [27,57]. More efforts are needed to identify these RUNX3 E3 ligases. Since many E3 ligases are dysregulated in cancer with altered activity and expression level [96], an inverse expression correlation between RUNX3 E3 ligases and RUNX3 itself might exist in certain cancers. Due to the differential expression patterns of E3 ligases in cancers [96], it is likely that E3 ligases involved in the ubiquitination of RUNX3 varies in different cancers. 

RUNX3 undergoes a variety of post-translational modifications, including phosphorylation, acetylation and ubiquitination, and these different modifications contribute to regulating the stability of RUNX3 to some extent, as discussed above. Importantly, post-translational modifications also play a key role in RUNX3-mediated ubiquitination and degradation of oncogenic proteins and the activation of certain E3 ligases [96]. Since many of these post-translational modifications are catalyzed by similar enzymes, it remains an important question to determine how the tumor suppressor function of RUNX3 is coordinately regulated by the same type of modification on RUNX3 itself, its targeted proteins, and the UPS. Crosstalk between different post-translational modifications could also be critical for this regulation. 

Targeted therapy and immunotherapy are two advanced strategies for cancer treatment. While restoration of RUNX3 is able to suppress proliferation of various cancer cells in vitro, selectively targeting RUNX3 for its reactivation in human cancer cells remains a great challenge. Furthermore, the UPS has also been considered a potential target for cancer immunotherapy. For example, MDM2 inhibitors have been shown to sensitize cancer cells to T-cell-mediated killing, or synergies with PD1 blockade in a mouse model of immunotherapy [97,98]. Considering that MDM2 is an E3 ligase for RUNX3, it would be of great interest and importance to investigate whether reactivation of RUNX3 in cancer cells would increase the efficacy of immunotherapy in the future.

## Figures and Tables

**Figure 1 cells-12-00717-f001:**
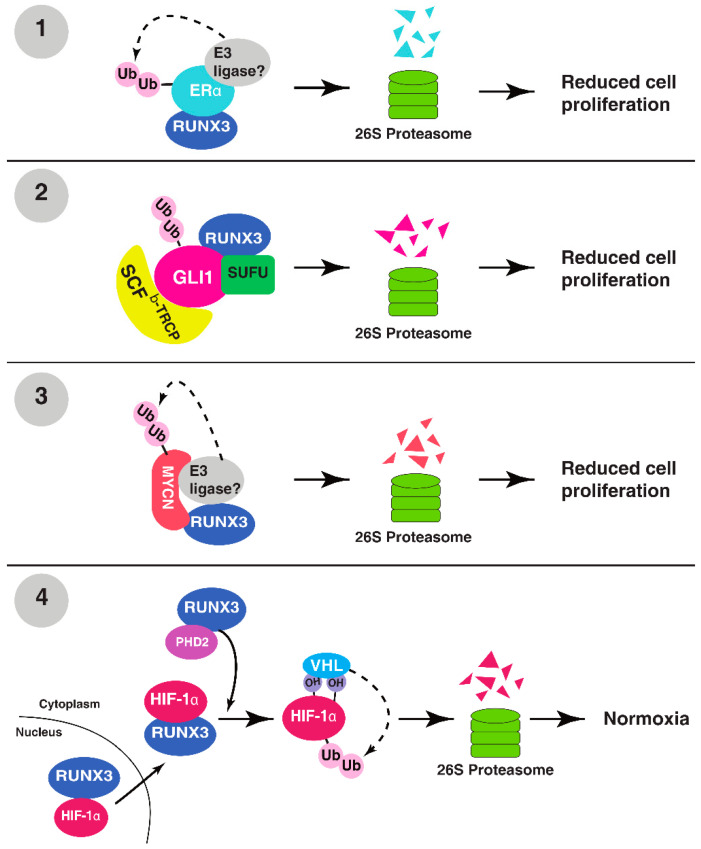
RUNX3 targets oncogenic proteins for ubiquitination and degradation. (**1**) RUNX3 binds to the hinge region of ERα, destabilizing it for proteasomal degradation. (**2**) RUNX3 binds to GLI1–SUFU complex and recruits SCF^β-Trcp^ for proteasomal degradation of GLI1. (**3**) RUNX3 binds to MYCN and recruits a yet unknown E3 ligase for the polyubiquitination and proteasomal degradation of MYCN. (**4**) RUNX3 binds to and transports HIF-1α into the cytoplasm, where RUNX3-bound PHD2 hydroxylates HIF-1α, promoting the recruitment of VHL for the ubiquitination and degradation of HIF-1α.

**Figure 2 cells-12-00717-f002:**
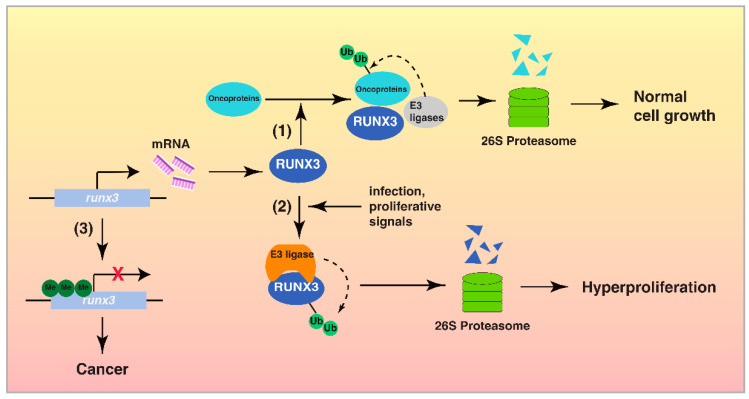
The interplay between RUNX3 and the UPS in cancer development. Step (1) In normal cells, RUNX3 is transcribed actively and functions as a tumor suppressor by its interaction with the UPS, to induce the proteasomal degradation of oncoproteins and control cell proliferation. Step (2) In response to certain stimuli, including *H. pylori* infection, RUNX3 is targeted by the UPS for degradation, leading to uncontrolled cell proliferation. Step (3) Hypermethylation of *Runx3* promoter results in the epigenetic silencing of RUNX3 and the development of cancer.

**Table 1 cells-12-00717-t001:** Factors or signals regulating the stability of RUNX3.

Regulators	Cancer/Cells	E3 Ligases	Mechanisms of Action	Functional Outcomes	References
HOTAIR	Gastric cancer	Mex3b	Decreased stability of RUNX3 via E3 ligase Mex3b	Promotes cancer cell metastasis	[49]
HAGLR	Treg cells	U/I	Destabilization of RUNX3 by direct binding	Development of dermatomyositis	[51]
PIN1	Breast cancer	U/I	Decreased RUNX3 stability by direct binding to the 4 pSer/Thr-Pro motifs of RUNX3	Promotes cell proliferation	[56]
JAB1/CSN5	Multiple cancers	U/I	Decreased RUNX3 stability by regulating its nuclear export and CK2α-mediated phosphorylation	Inhibits transcriptional activity of RUNX3	[61]
CagA	Gastric cancer	U/I	Decreased RUNX3 stability by binding to its PY motif via CagA’s WW domain	Inhibits transcriptional activity of RUNX3	[12,65]
K-ras-V12	Multiple cancers	MDM2	Increased RUNX3 stability by activating p14^ARF^, which binds to and inhibits MDM2, a E3 ligase for RUNX3	Increases transcriptional activity of RUNX3	[27]
TGF-β-p300	Multiple cancers	Smurfs	Increased RUNX3 stability by p300-mediated acetylation of the same lysine residues targeted by Smurfs for ubiquitination	Increases transcriptional activity of RUNX3	[57]
PTHrP-CDK4	Chondrocytes	U/I	Decreased RUNX3 stability by inducing cyclin D1/CDK4-mediated phosphorylation of S536	Promotes cell proliferation	[74]

U/I = unidentified.

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
