# Peer review of "RUNX3 Meets the Ubiquitin-Proteasome System in Cancer"

_cells, 2023, doi:10.3390/cells12050717_

Round 1

Reviewer 1 Report

The review manuscript is well written and organized. The topic is important to the RUNX3, protein ubiquitination/degradation and cancer fields.

Author Response

We thank the reviewer for his/her very positive comments.

Reviewer 2 Report

This review article from Toska et al. is overall focused and well-written. The authors has comprehensively covered the current knowledge of the tumor suppressor function of RUNX3 by facilitating the ubiquitin-dependent degradation of several oncogenic proteins as well as by serving as a target of the UPS in certain cancers. However, minor revisions are required to for acceptance:

1. In figure 1, the authors included the oncogenic proteins that undergo ubiquitin-dependent degradation mediated by RUNX3. The roles of RUNX3 are indicated as controlling NORMAL cell growth/response, which is inconsistent with the reports and the scope of this review- CANCER cell death. Authors should revise this figure.

2. Subsection 3.5 does not fit in Section 3, which can be moved to Conclusion and discussion or serve as an independent section.

3. If authors would like to include Section 4., starting with something like "in addition to its previously described role as a tumor suppressor" may help the logic.

4. A recent paper identified a peptide that target RUNX3 phosphorylation (PMID 34253860). The authors may also include this in the discussion.

5. A concluding figure of the regulation between RUNX3 and UPS in cancer may be useful to highlight the scope of this review. 

Author Response

  1. In figure 1, the authors included the oncogenic proteins that undergo ubiquitin-dependent degradation mediated by RUNX3. The roles of RUNX3 are indicated as controlling NORMAL cell growth/response, which is inconsistent with the reports and the scope of this review- CANCER cell death. Authors should revise this figure.

Thank you for recognizing this inconsistency. The figure has been revised accordingly.

  1. Subsection 3.5 does not fit in Section 3, which can be moved to Conclusion and discussion or serve as an independent section.

Subsection 3.5 was converted to an independent section: “Regulation of RUNX3 Stability by Other Post-Translational Modifications”

  1. If authors would like to include Section 4., starting with something like "in addition to its previously described role as a tumor suppressor" may help the logic.
    We have removed Section 4 because it is inconsistent with the scope and logical flow of the review in characterizing the interaction of tumor suppressive RUNX3 and the UPS.
  2. A recent paper identified a peptide that target RUNX3 phosphorylation (PMID 34253860). The authors may also include this in the discussion.

Thank you for suggesting this important addition. This paper has been included in the discussion of RUNX3 as a therapeutic target and its possible clinical applications in Section 5.

  1. A concluding figure of the regulation between RUNX3 and UPS in cancer may be useful to highlight the scope of this review.

A concluding figure has been added to summarize the tumor suppressive role of RUNX3-UPS interaction in the development of cancer, which represents the scope of the review.

Reviewer 3 Report

This work entitled “RUNX3 Meets the Ubiquitin-Proteasome System in Cancer” by Albano Toska et al., have revised the role of the transcription factor RUNX3 in Ubiquitination and protein degradation system in Cancer. Cancer cells are generally distinguished by one or both of two homeostatic instabilities: they can have uncontrollable cellular proliferation, and/or a distinct lack of apoptosis. The authors presented well the RUNX3 role associated with the proteasome involvement in regulatory pathways within the cell, including Ubiquitination and Degradation of Oncogenic Proteins, and Inactivation of RUNX3  itself. However, the following minor changes are suggested to the present review:

1. The topic “Oncogenic RUNX3 and Ubiquitination” lacks better explanation linking the mechanisms of how RUNX3 switches from tumor suppressive to oncogenic activity; this is of clinical relevance with implications for cancer detection and prognosis. If possible, that should better addressed.

2. The authors should also explore the topic ‘targeting the ubiquitin-proteosome system and its implications for cancer therapy’, including the relevance of RUNX3 for it.

3. Also, it would be interesting see a bit more of latest novel anti-cancer therapeutical tools regarding the via proteasome inhibitors, including clinical studies, in ‘Concluding Remarks and Prospective’ section of the review.

4. Finally, to conclude, the authors should link better the findings of  RUNX3-associated role UPS in cancer and its clinical relevance and implications for cancer detection and prognosis.

Author Response

  1. The topic “Oncogenic RUNX3 and Ubiquitination” lacks better explanation linking the mechanisms of how RUNX3switches from tumor suppressive to oncogenic activity; this is of clinical relevance with implications for cancer detection and prognosis. If possible, that should better addressed.
  • Thank you for acknowledging this inconsistency. We have removed the section on oncogenic RUNX3 because it is outside the scope of this review on tumor suppressive RUNX3 and the UPS. The therapeutic potential of targeting RUNX3 and the UPS was also further explored in new Section 5.
  1. The authors should also explore the topic ‘targeting the ubiquitin-proteosome system and its implications for cancer therapy’, including the relevance of RUNX3 for it.
  • We have added a new section 5 to discuss targeting RUNX3-UPS interplay in cancer therapy.
  1. Also, it would be interesting see a bit more of latest novel anti-cancer therapeutical tools regarding the via proteasome inhibitors, including clinical studies, in ‘Concluding Remarks and Prospective’ section of the review.
  • Clinical studies of a novel RUNX3-targeted therapeutic candidate as well as the latest UPS-targeted therapies that may apply to maintaining RUNX3 expression have been included in Section 5. Implications of RUNX3 on immunotherapies are also discussed in Section 6.
  1. Finally, to conclude, the authors should link better the findings of RUNX3-associated role UPS in cancer and its clinical relevance and implications for cancer detection and prognosis.
  • We expatiated on the intersection between RUNX3-UPS and clinical applications in Section 5. We discuss epigenetic as well as UPS-targeting approaches as possible future therapeutics in the context of rescuing RUNX3 expression in cancer.